# Joint Models to Predict Dairy Cow Survival from Sensor Data Recorded during the First Lactation

**DOI:** 10.3390/ani12243494

**Published:** 2022-12-10

**Authors:** Giovanna Ranzato, Ines Adriaens, Isabella Lora, Ben Aernouts, Jonathan Statham, Danila Azzolina, Dyan Meuwissen, Ilaria Prosepe, Ali Zidi, Giulio Cozzi

**Affiliations:** 1Department of Animal Medicine, Production and Health (MAPS), University of Padova, Viale dell’Università 16, 35020 Legnaro, Italy; 2Division of Animal and Human Health Engineering, Department of Biosystems, KU Leuven, Campus Geel, Kleinhoefstraat 4, 2440 Geel, Belgium; 3Animal Breeding and Genomics, Wageningen University and Research, P.O. Box 338, 6700 AH Wageningen, The Netherlands; 4RAFT Solutions Ltd., Sunley Raynes Farm, Galphay Road, Ripon HG4 3AJ, UK; 5Harper & Keele Veterinary School, 2 Church Bank, Keele, Newcastle ST5 5NS, UK; 6Department of Environmental and Preventive Sciences, University of Ferrara, Corso Ercole I d’Este 32, 44121 Ferrara, Italy; 7Unit of Biostatistics, Epidemiology and Preventive Sciences, Department of Cardiac, Thoracic, Vascular Sciences and Public Health, University of Padova, Via L. Loredan 18, 35131 Padova, Italy

**Keywords:** dairy cow, sensor data, survival, joint model, decision support tool

## Abstract

**Simple Summary:**

Dairy farmers would benefit from a decision support tool that predicts each cow’s probability of survival to future lactations. Based on this output, they might optimize herd breeding decisions by selecting the cows that better cope with the existing housing and management conditions of their own farm. This work explored the accuracy of a novel statistical technique to obtain predictions of cows’ probabilities of survival to the second and third lactations, starting from sensor data of daily milk yield, body weight, and rumination time automatically recorded during different stages of the cows’ first lactation. Data from six different dairy farms were individually analyzed; in almost all the scenarios, the error associated with the obtained survival predictions was low. The explored decision model applied to the dairy cattle sector revealed good potentialities.

**Abstract:**

Early predictions of cows’ probability of survival to different lactations would help farmers in making successful management and breeding decisions. For this purpose, this research explored the adoption of joint models for longitudinal and survival data in the dairy field. An algorithm jointly modelled daily first-lactation sensor data (milk yield, body weight, rumination time) and survival data (i.e., time to culling) from 6 Holstein dairy farms. The algorithm was set to predict survival to the beginning of the second and third lactations (i.e., second and third calving) from sensor observations of the first 60, 150, and 240 days in milk of cows’ first lactation. Using 3-time-repeated 3-fold cross-validation, the performance was evaluated in terms of Area Under the Curve and expected error of prediction. Across the different scenarios and farms, the former varied between 45% and 76%, while the latter was between 3.5% and 26%. Significant results were obtained in terms of expected error of prediction, meaning that the method provided survival probabilities in line with the observed events in the datasets (i.e., culling). Furthermore, the performances were stable among farms. These features may justify further research on the use of joint models to predict the survival of dairy cattle.

## 1. Introduction

Cow survival is a complex trait that depends on multiple factors, such as milk production, fertility, health, and farm management conditions [1]. If survival is computed from the day of the first calving, it coincides with the productive life of the animal, which represents a very important trait in the dairy practice [2]. Typically, cows with longer productive lives are more resilient, exhibiting good productive and reproductive performances and having few health problems that they overcome rapidly [3]. Nowadays, the average cow productive life ranges from 2.5 to 3.5 lactations [4,5], while a dairy cow is biologically capable of a life span up to 20 years [6]. Additionally, the research by Bach [7] reported a decline in survival rates of first-parity cows. When dairy cows do not manage to survive beyond the first lactation, the rearing costs are not paid back; cows start making profit for the farmer only during the second lactation, reaching the full production potential during the third lactation [8]. Moreover, Grandl et al. [9] showed that cows that do not complete the first lactation perform particularly unfavorably with regard to their greenhouse gas emissions per unit of produced milk. Moreover, from an ethical perspective, short longevity is typically an indicator of poor animal welfare, being a sign of impaired biological functions and health conditions [10].

Dairy farmers would benefit from a tool able to provide information about the future prospect of the first-parity cows in their herds. Based on survival predictions at farm level, they could select the ones that better cope with the existing housing and management conditions, optimizing culling decisions and breeding schemes. To date, no decision support tools have been implemented to help farmers in selecting the cows that are more likely to thrive in their own farm environment. Nowadays, some possibilities can arise from the great amount of information provided by the increasing number of sensor systems operating on many dairy farms [11,12]. These new technologies provide a constant flow of high-frequency repeated measures of parameters, such as milk yield and quality (e.g., somatic cell count) or a cow’s activity (e.g., locomotion and rumination), which can reflect changes in the physiological and health status of the animal [13,14]. These measurements can be used to predict cow survivability using new statistical methods. These methods are based on the joint modelling of longitudinal and time-to-event data [15]. Joint models are used in the field of biomedicine to predict patients’ survival probabilities based on temporal trajectories of disease-specific biomarkers and to discriminate between patients with a low or high risk of mortality. These models are versatile, being easily adapted to different recording periods of longitudinal data, time points of survival prediction, and variables to be used in the models. Furthermore, joint models avoid deriving biologically meaningful proxies from time-series data, since they directly estimate the information provided by the raw (nearly unprocessed) longitudinal data.

The aim of the present study was to explore the adoption of a joint model that used first-lactation longitudinal sensor data of milk yield (MY), body weight (BW), and rumination time (RUM) to predict cows’ survival to subsequent lactations.

## 2. Materials and Methods

### 2.1. Data

Data were retrieved from 6 Holstein dairy farms (3 British, 2 Belgian, and 1 Italian) equipped with automatic milking systems (AMS) of Lely Industries (Lely Industries N.V., Maasluis, The Netherlands). Farms were selected based on data availability and on farmers’ willingness to participate in the study. Daily records of individual cow MY, BW, and RUM were collected from the AMS database, to be used as potential indicators of lactating cows’ health status [16,17] and, therefore, as information possibly related to their survival. Dates of cows’ birth, calving, and culling were also retrieved from the farm databases. The time period covered by all the datasets varied between 2013 and 2020. Descriptive information for each farm is reported in Table 1.

### 2.2. Data Processing

Data processing and analysis were performed with RStudio software (R version 4.1.2; RStudio PBC, Boston (MA), USA) and equally conducted for each dataset (i.e., farm).

The survival time (T) of each cow was computed as the number of days between the first calving and the culling, coinciding with the productive life of the animal. Culling dates were derived from the last date on which milk production was registered. If no culling date was available, the cow was considered still alive at the final date of the dataset (i.e., censored), and the survival time was computed as the difference in days between the final date and the date of the first calving; the cow was removed if she had not yet completed the first lactation at the end date of the dataset. The age at first calving (AFC) of each cow was expressed as a 3-category variable: ‘low’ if it was below the first quartile of herd AFC, ‘medium’ if it was within the interquartile range, and ‘high’ if it was above the third quartile. The season of the data recording period (SEAS) was transformed into a binary variable: ‘warm’ if between April and October, ‘cold’ if otherwise.

Individual cow raw sensor data of MY, BW, and RUM recorded during first lactations were used in the study. Farm databases provided daily MY and BW in kilograms, while RUM data consisted of 2-hourly measures that were summed into single daily records expressed in minutes. According to Adriaens et al. [3], values of each sensor variable that fell outside of 3 standard deviations (SD) from the respective herd means were treated as outliers and removed from the dataset, except when they were present more than 30 times for the same cow. The rationale was to clean the dataset of errors in the data recording while keeping the information related to possible real disturbances (such as diseases). This was assuming a cow had an actual ‘abnormal behavior’ when outliers characterized a total of at least 30 days of the whole lactation time. Table 2 reports means and SDs of daily MY, BW, and RUM for every farm.

All the cows culled before 50 days in milk (DIM) of the first lactation (i.e., T < 50) were deleted from the dataset to examine only animals with a reasonable amount of sensor observations. Moreover, we considered first-lactation sensor measurements in the interval 5–305 DIM; the starting point was set at 5 DIM to avoid missing data associated with the very first days after calving, while the maximum observed time was set at 305 DIM, as it is the standard lactation length used for genetic evaluations in cattle [18]. After the data-filtering and cleaning procedures, a cow was removed from the dataset if she remained with less than 90% daily observations with respect to the first-lactation length (maximum 305 DIM).

### 2.3. Algorithm Development

An algorithm based on multivariate joint modelling of longitudinal and time-to-event data was built to predict cow survival from raw daily data of MY, BW, and RUM recorded during the first lactation; ‘multivariate’ refers to the presence of 3 longitudinal variables to be modelled simultaneously.

The joint modelling technique has been recently studied by Rizopoulos [15]. It consists of two steps: (i) description of the evolution of the longitudinal variable over time using a (generalized) linear mixed model [19] and (ii) estimation of the survival probabilities using the estimated evolution within a survival Cox model [20]. Assuming i=1,…,n is the statistical unit (e.g., patient) and k=1,…,K identifies the different longitudinal outcomes, the evolution over time t of each outcome $y_{ik}$ can be described by the following linear mixed model:(1)$$\{\begin{array}{l} y_{ik}\left(t\right)=x_{i}^{Τ}\left(t\right)\beta_{k}+z_{i}^{Τ}\left(t\right)b_{ik}+\varepsilon_{ik}\left(t\right)\\ b_{ik}\sim N\left(0,D_{k}\right),\;\;\varepsilon_{ik}\left(t\right)\sim N\left(0,\sigma_{k}^{2}\right) \end{array},$$
where $x_{i}$ are the predictors associated with the fixed effects $\beta_{k}$, $z_{i}$ are the predictors associated with the random effects $b_{ik}$, and $\varepsilon_{ik}$ is the error term. Both the vector of the random effects and the vector of the errors have a normal distribution. The correlation between the different longitudinal variables $y_{ik}$ is then captured by setting a multivariate normal distribution for the random effects $b_{i}={\left(b_{i1},\ldots ,b_{iK}\right)}^{T}\;\sim \;N\left(0,D\right)$. Assuming $m_{ik}\left(t\right)=x_{i}^{Τ}\left(t\right)\beta_{k}+z_{i}^{Τ}\left(t\right)b_{ik}$ is the ‘true’ value of each outcome at time t, we can define the following multivariate joint model (i.e., Cox hazard model containing the evolution processes of the longitudinal outcomes):(2)$$h_{i}\left(t|{ℳ}_{i1}\left(t\right),\ldots ,{ℳ}_{iK}\left(t\right)\right)=h_{0}\left(t\right)\exp\left(\gamma^{Τ}\omega_{i}+\overset{K}{\underset{k=1}{\sum }}\alpha_{k}m_{ik}\left(t\right)\right).$$

The equation ${ℳ}_{ik}\left(t\right)=\{m_{ik}\left(s\right),$ 0 ≤s≤t} represents the longitudinal history of $m_{ik}$ until t, where $h_{0}\left(t\right)$ is the baseline hazard function at time t, $\alpha_{k}$ measures the association between $m_{ik}$ and the risk of an event, and $\omega_{i}$ are baseline variables. The joint estimation process is carried out with a Markov Chain Monte Carlo algorithm [21].

According to this theoretical approach, in the present study, the K longitudinal variables were represented by first-lactation daily sensor data: $\left(MY_{i}\left(t\right),BW_{i}\left(t\right),RUM_{i}\left(t\right)\right)=\left(y_{i1}\left(t\right),y_{i2}\left(t\right),y_{i3}\left(t\right)\right)$. The evolution of each $y_{ik},\;k=1,\;2,\;3$ over t was described by the following linear mixed model:(3)$$y_{ik}\left(t\right)=\beta_{0k}+\beta_{1k}\mathrm{ns}\left(t\right)+\beta_{2k}AFC_{i}+\beta_{3k}SEAS_{i}\left(t\right)+b_{i0k}+b_{i1k}\mathrm{ns}\left(t\right)+\varepsilon_{ik},\;\;\;\;\;\;\;\;\;\;\;\;\;\;\;\;\;\;\;\;\;\;\;\;\;\;\;\;\;\;\;\;\;\;\;\;\;\;\;\;\;\;\;\;\;\;\;\;\;\;\;\;\;\;\;\;\;\;\;\;\;\;\;\;\;\;\;\;\;\;\;\;\;\;\;\;\;\;\;\;\;\;\;\;\;\;\;\;\;\;\;\;\;\;\;\;\;\;\;\;\;\;\;\;\;\;\;\;\;\;\;\;\;\;i=1,\ldots ,n,$$
where n was the number of cows in the dataset. The fixed effects $\beta_{k}={\left(\beta_{0k},\beta_{1k},\beta_{2k},\beta_{3k}\right)}^{T}$ were respectively associated with the intercept of the model, the time t expressed as DIM (5 ≤ DIM ≤ 305), the cow’s AFC, and SEAS at t. More specifically, the time was modelled with a natural cubic spline (ns). The spline was set to have one knot at the median DIM of the dataset (resulting in 2 different cubic sub-polynomials) when RUM was the longitudinal outcome. For MY and BW, the splines were set to have 3 knots at the 3 quartiles of DIM of the dataset (resulting in 4 different cubic sub-polynomials) to capture the well-defined shapes of the trend of these two traits over an entire lactation [22]. The random effects $b_{ik}={\left(b_{i0k},b_{i1k}\right)}^{T}$ were respectively associated with the cow-specific intercept and the cow-specific time slope. The random intercept was necessary to capture the variation of the parameters of the i^th^ animal from those in the dataset, while the random slope allowed the evolution in time described by ns(t) to be different from one cow to another. The correlation between $y_{i1}$, $y_{i2}$, and $y_{i3}$ was captured using $b_{i}={\left(b_{i01},b_{i11},b_{i02},b_{i12},b_{i03},b_{i13}\right)}^{T}\;\sim \;N\left(0,D\right)$ with unstructured covariance matrix D. Assuming $m_{ik}\left(t\right)=\beta_{0k}+\beta_{1k}\mathrm{ns}\left(t\right)+\beta_{2k}AFC_{i}+\beta_{3k}SEAS_{i}\left(t\right)+b_{i0k}+b_{i1k}\mathrm{ns}\left(t\right)$ (i.e., the sensor value without error), we defined the following multivariate joint model:(4)$$h_{i}\left(t|{ℳ}_{i1}\left(t\right),{ℳ}_{i2}\left(t\right),{ℳ}_{i3}\left(t\right)\right)=h_{0}\left(t\right)\exp\left(\gamma_{1}AFC_{i}+\alpha_{1}m_{i1}\left(t\right)+\alpha_{2}m_{i2}\left(t\right)+\alpha_{3}m_{i3}\left(t\right)\right),$$
where the event was represented by ‘the cow was culled by the last date of the dataset’. The risk of being culled at t could then be associated with the first-lactation levels of MY, BW, and RUM at t, adjusted by the animal’s AFC (baseline variable).

We supposed it was more likely that the risk of being culled at t could be associated with the slopes of the trajectories of the sensor variables at t, and not with their current values as in the previous model specification (4). In this way, the joint estimation process could identify fluctuations in the sensor measurements resulting from possible disturbances (such as diseases) and examine their relationship with the cow at risk of being culled. An illustrative example is reported in Figure 1 for the MY variable related to one cow; the lactation curve deviates from the typical lactation curve of dairy cattle, and this deviation is captured by the slope. The final model used in the study was then expressed by the following equation:(5)$$h_{i}\left(t|{ℳ}_{i1}\left(t\right),{ℳ}_{i2}\left(t\right),{ℳ}_{i3}\left(t\right)\right)=h_{0}\left(t\right)\exp\left(\gamma_{1}AFC_{i}+\alpha_{1}m_{i1}^{\prime }\left(t\right)+\alpha_{2}m_{i2}^{\prime }\left(t\right)+\alpha_{3}m_{i3}^{\prime }\left(t\right)\right),$$
where $m_{ik}^{\prime }\left(t\right)=\frac{d}{dt}\left\{\beta_{0k}+\beta_{1k}\mathrm{ns}\left(t\right)+\beta_{2k}AFC_{i}+\beta_{3k}SEAS_{i}\left(t\right)+b_{i0k}+b_{i1k}\mathrm{ns}\left(t\right)\right\}$ was the time-dependent slope of the sensor variable k, k=1,2,3, for cow i (i.e., the first derivative of $m_{ik}\left(t\right)$).

The modelling was carried out with R package ‘JMbayes2’ (version 0.2-0, published 2022-02-10; [23]).

### 2.4. Algorithm Evaluation

To evaluate the performance of the algorithm, avoiding data underfitting or overfitting, repeated 3-fold cross-validation (CV) was used in every farm dataset. All the cows of the dataset were randomly partitioned into 3 groups of similar sizes; then 2 of these groups were used to train the model, and the third group was used to test it. This operation was repeated 3 times, rotating the groups [24]. The same procedure was again repeated 3 times in total, and the mean performance across all folds from all runs was reported (i.e., mean of 9 single results per farm).

During the training, 67% of the animals in the dataset were used to fit the joint model. The model was trained on sensor data recorded during 5–305 DIM of the first lactation and on the cows’ observed survival times, and the effect of the trajectory of each sensor variable on the risk of being culled was estimated. The testing used 33% of the cows to evaluate the prediction performance. The model accuracy in predicting cow survival was tested under 6 different scenarios: 2 different time points of prediction (i.e., second and third calving) from sensor data recorded during 3 different observation periods of the cow’s first lactation (i.e., 60, 150, and 240 DIM). Survival was therefore predicted at $t_{1}$ = ‘second calving’ and $t_{2}$ = ‘third calving’, respectively, and estimated as once and twice the average calving interval (in days) after the date of the first calving for all the cows of the farm. A summary of the values of $t_{1}$ and $t_{2}$, along with the number of cows that were culled before them, is reported for each dataset in Table 1.

Given that $\Upsilon_{i}\left(v\right)=\left\{y_{ik}\left(s\right),\;5\leq s\leq v,\;v=60,\;150,\;240,\;k=1,2,3\right\}$ represented the available first-lactation sensor measurements for a ‘new’ cow i of the testing set that had provided MY, BW, and RUM values up to v, individualized predictions of the survival probabilities up to $t_{j},\;j=1,2$, for cow i was obtained by estimating
(6)$$\pi_{i}\left(u|v\right)=\mathrm{Pr}\left\{T_{i}\geq u|T_{i}>v,\;\;\Upsilon_{i}\left(v\right),\;\;ℛ\right\}$$
where $v<u\leq t_{j}$, and ℛ denoted the sample on which the model was fitted (i.e., the training set). Providing measurements up to time v implied that the cow was still alive at v (i.e., $T_{i}>v$); in every testing set, only the animals that had survived at least up to 240 DIM (i.e., the maximum period of days considered) were then examined. Assuming a specific threshold value c∈(0,1) (here c= 0.5), cow i was finally predicted ‘culled at $t_{j}$’,  j=1,2, if $\pi_{i}\left(t_{j}|v\right)\leq c$. Two measures of prediction accuracy were accordingly computed based on the value of $\pi_{i}\left(t_{j}|v\right)$: the Area Under the Curve (AUC) [25] and the expected error of prediction (PE, Prediction Error) [26]. The AUC measured the ability of the model to distinguish between the classes ‘culled at $t_{j}$’ and ‘still on farm at $t_{j}$’, representing a measure of its discrimination capability (0 ≤ AUC ≤ 1). The PE measured the accuracy of the obtained survival predictions by computing the average squared distance between the survival status (i.e., culled or alive) and the predicted survival probability, making it a measure of the calibration capability of the model (0 ≤ PE ≤ 1). The higher the AUC, the better the model performed at predicting the cows that were culled within $t_{j}$ as actually ‘culled at $t_{j}$’ and the cows that were still on the farm at $t_{j}$ as ‘still on farm at $t_{j}$’; the lower the PE, the more the survival predictions were aligned with the observed events (i.e., culling) within $t_{j}$.

## 3. Results

To clearly illustrate the algorithm training phase, Table 3 shows the output of the fitting obtained in one training set (148 cows; 40,995 observations) of the repeated CV procedure for one of the available farms. In this case, the longitudinal modelling process highlighted the presence of between-cow variability, expressed by the estimated SD of the random effects for the three sensor outcomes (MY, BW, and RUM). Focusing on the survival process, the slope of RUM ($\alpha_{3}$) was negatively associated with the risk of being culled, keeping all other variables constant. This implied that a lower value of the slope was associated with poorer survival probability.

The mean AUC and mean PE over the 9 CV runs (3 × 3 folds) are reported in Table 4. For some farms (‘Belgian 2’ and ‘British 2’), there were no culling events registered within $t_{1}$ (i.e., second calving) in any testing set of the CV procedure; therefore, the performance metrics at $t_{1}$ could not be estimated. To determine the significance of the performance metrics over 0.50 for AUC and below 0.25 for PE (i.e., algorithm performing random guessing between ‘culled’ and ‘alive’ [27]), we constructed a 95% confidence interval using the mean and the standard deviation obtained from the 9 CV repetitions for each farm in each scenario. The PE values were always significantly lower than 0.25 at $t_{1}$ (i.e., second calving) and, in most cases, at $t_{2}$ (i.e., third calving) (Table 4); PE was generally low at $t_{1}$, suggesting that the model accurately predicted the events within the second calving. The AUC was significantly higher than 0.50 only in a few cases, both for the predictions at $t_{1}$ and at $t_{2}$, remaining generally close to 0.50 (Table 4). Only one farm reported an average AUC of 0.76 at $t_{1}$, with 240 DIM of first lactation sensor observations to obtain predictions. It is worth noting that this was the dataset that, across training sets, had the highest number of significant associations between the sensor variables and survival, meaning that the sensor information was, in this case, particularly useful for predicting the animals’ survival. These results revealed that the algorithm had a good calibration capability (PE), but the same did not apply for its discrimination capability (AUC). However, the average model performance metrics tended to improve with more days of longitudinal information (i.e., 240 vs. 150 vs. 60 DIM) and when predicting survival at closer endpoints (i.e., $t_{1}$ vs. $t_{2}$). Furthermore, the results from Levene’s tests [28] conducted in each scenario to verify the homogeneity of variances of the AUC and PE among farms revealed that the performance metrics of the algorithm were stable. Only AUC values estimated at the third calving with 60 or 150 DIM information had different variances among farms (respectively, *p* = 0.01 and *p* = 0.02).

Figure 2 represents a possible output of the algorithm, obtained by a farmer for a ‘new’ cow of his/her herd. The farmer may decide to keep this cow for breeding purposes, given that at 150 DIM of the first lactation, she has a predicted probability of surviving to the second calving equal to 90%.

## 4. Discussion

This study explored the possibility of using joint models to predict dairy cow survival at different lactations, starting from raw daily sensor data recorded on-farm during different (early) stages of the first lactation. The algorithm implemented in this work could represent the basis for a prognostic model-based tool able to inform farmers of the future prospect of each first-parity cow in their herds. This may be very useful in the early adjustment of herd breeding and management decisions, improving farm efficiency and sustainability; farmers could, for instance, optimize the use of dairy sexed and beef semen or decide whether to give another chance to those cows that are not pregnant after two or three inseminations.

The performances of the algorithm were compared with the results of the few similar studies dealing with dairy cow survival predictions and/or longitudinal sensor data extracted from AMS. Van der Heide [29] predicted survival to the second lactation using breeding and phenotypic variables from different moments in the heifer’s life. The authors compared three different machine-learning methods for many performance metrics, including AUC. Average AUC was 0.67 when using the information available at 6 weeks post-first calving (i.e., 40–50 DIM) and 0.68 when using the information at 200 DIM. The performance of these models was then higher compared to our average results (AUC = 0.54 ± 0.05 at second calving using 60 DIM, and AUC = 0.64 ± 0.09 at second calving using 240 DIM; mean ± SD), but their ability to correctly identify non-surviving animals was very low (average positive predictive value of 0.17). The same authors tried to improve these performances by using ensemble-learning approaches [1], which were expected to have better performances and more robustness, but the results remained quite poor (average positive predictive value of 0.20). Adriaens et al. [3] studied the possibility of predicting lifetime resilience and the productive life of dairy cows starting from sensor-derived proxies of first-parity daily sensor data, obtaining a mean classification performance (‘low’ vs. ‘medium’ vs. ‘high’ lifetime resilience rank) of 47 ± 8% (± SD), when using milk yield features alone, and of 56 ± 12% when using lactation and activity features together. Ouweltjes et al. [17] assessed the performance of different models that included milk yield, body weight, rumination, and activity sensor data of cows in first lactation to predict lifetime resilience. Model performances, expressed in the percentage of correctly classified cows (‘low’ vs. ‘medium’ vs. ‘high’ lifetime resilience rank), ranged between 45 ± 8% (mean ± SD) and 51 ± 6%.

The results of this research, in line with the results of the other works, confirm that cow survival is a complex trait, difficult to accurately predict [1]. It indeed combines several different factors, such as fertility, health, milk production, farm management, and environmental conditions [30]. With the only information at our disposal (i.e., AFC, SEAS, MY, BW, RUM), we could capture a small portion of these aspects; for instance, having information on disease occurrence would have likely improved the predictive performance of the algorithm. Furthermore, to build an algorithm applicable to all the farms with MY, BW, and RUM data from AMS, we had to ignore all the local and evidence-based farm management rules, which are particularly relevant when developing decision support tools for dairy farms [3].

We identified two main strengths of the methodology presented in this study. First, in contrast to other works with similar research goals [3,31], the present joint modelling approach has the practical advantage of not requiring the translation of sensor time-series data into biologically meaningful sensor features. Using raw sensor data to obtain longevity predictions avoids proper feature definition and a lot of pre-processing (thus reducing the chance of errors) and provides at least the same performance as models with pre-processed data, as demonstrated by Ouweltjes et al. [17]. Second, joint models have the advantage of being very flexible; they allow for the dynamic update of predicted survival probabilities as additional longitudinal data are recorded, as well as for the easy change of the final time point of prediction based on the target the user wants to test. These features may justify future research to improve the current performance within a farm. The model, for instance, could be tested by including additional variables from automated technologies (e.g., cow activity, somatic cell count) or cows’ additional information from other sources (e.g., test days, health records).

## 5. Conclusions

This study explored the potential of using joint models for longitudinal and time-to-event data to predict dairy cow survival at different lactations from raw sensor data recorded during different stages of the cow’s first lactation. The algorithm tested in this study had a modest performance in terms of discrimination accuracy (Area Under the Curve) but good results in terms of calibration accuracy (expected error of prediction), as well as good repeatability across different farms. The interesting opportunities that joint models offer in applicability and flexibility should justify further research in the attempt to improve the overall predictive accuracy in the dairy field.

## Figures and Tables

**Figure 1 animals-12-03494-f001:**
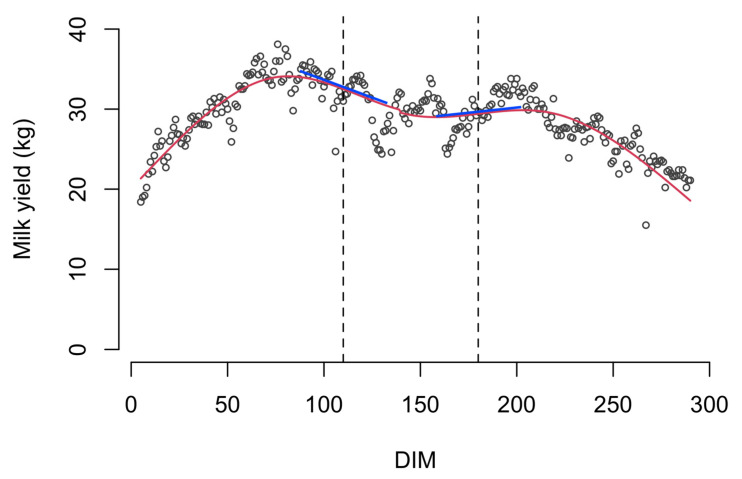
Tangent (blue) lines to the estimated milk yield trajectory (red curve) at time points t = 110 DIM and t = 180 DIM for one cow (5 ≤ DIM ≤ 290) of one farm randomly chosen. The joint model examines the association between the slope of the tangent line at t and the risk of being culled at t.

**Figure 2 animals-12-03494-f002:**
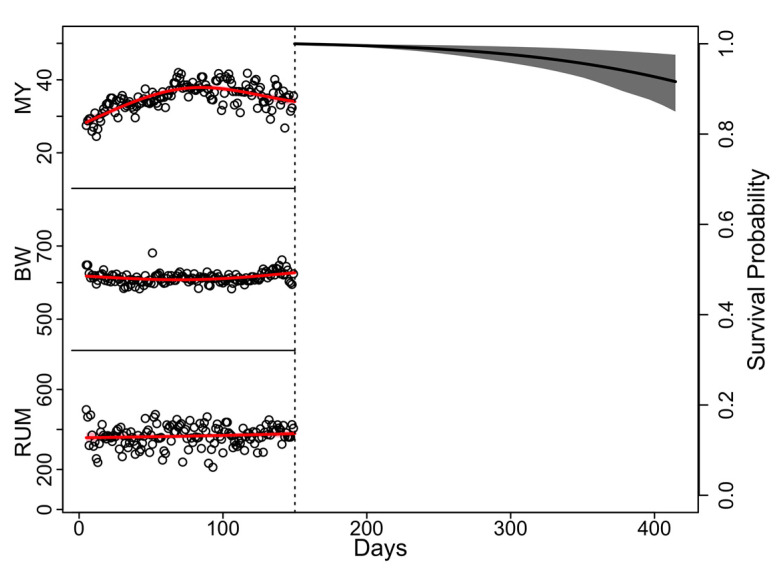
Predicted survival function from 150 DIM of the first lactation (dotted line) to the second calving (414 days post-first calving) for one cow of one farm, randomly chosen. The multivariate joint model estimates the evolutions of the sensor data (MY = Milk yield (kg), BW = Body weight (kg), RUM = Rumination time (min)) until 150 DIM of the first lactation (red curves) and, based on those, predicts the survival function until the second calving (black curve).

**Table 1 animals-12-03494-t001:** Overview of the available datasets.

Farm	Time Period	Cows (n)	$t_{1}$ ^1^	$\mathrm{Culled}\;\mathrm{before}\;t_{1}$	$t_{2}$ ^2^	$\mathrm{Culled}\;\mathrm{before}\;t_{2}$
Italian	2014–2020	98	414	12%	828	41%
Belgian 1	2014–2020	169	422	18%	843	34%
Belgian 2	2013–2019	182	397	9%	793	21%
British 1	2013–2019	266	384	9%	768	24%
British 2	2013–2019	101	402	11%	805	26%
British 3	2013–2019	226	400	6%	799	17%

^1^ Average number of days between the first and second calving; ^2^ average number of days between the first and third calving.

**Table 2 animals-12-03494-t002:** Means and standard deviations (SDs) of the recorded sensor data.

Farm	MY ^1^		BW ^2^		RUM ^3^	
	Mean	SD	Mean	SD	Mean	SD
Italian	32.8	6.60	595	62.1	458	94.2
Belgian 1	27.5	6.33	530	65.9	470	98.0
Belgian 2	32.4	6.57	548	109	487	126
British 1	31.9	8.01	634	65.1	491	102
British 2	24.2	6.11	567	60.0	500	120
British 3	33.9	6.84	578	59.0	484	125

^1^ Milk yield (kg/d); ^2^ body weight (kg/d); ^3^ rumination time (min/d).

**Table 3 animals-12-03494-t003:** Output from the fitted multivariate joint model in one dataset.

**Survival Outcome**						
**Parameters**	**Coeff. ^1^**				** *p* ^2^ **	
$\gamma_{1,1}$ (AFC ^3^ medium)	0.997				*	
$\gamma_{1,2}$ (AFC high)	1.35				*	
$\alpha_{1}$ (slope MY ^4^)	0.005				n.s.	
$\alpha_{2}$ (slope BW ^5^)	0.178				n.s.	
$\alpha_{3}$ (slope RUM ^6^)	−0.633				**	
**Longitudinal Outcomes**						
**Parameters**	** MY (k=1) **		** BW (k=2) **		** RUM (k=3) **	
** *Fixed* **	**coeff.**	** *p* **	**coeff.**	** *p* **	**coeff.**	** *p* **
$\beta_{0k}$ (intercept)	28.0	***	469	***	472	***
$\beta_{1,1k}$ (ns(DIM) 1) ^7^	6.42	***	95.2	***	−18.6	**
$\beta_{1,2k}$ (ns(DIM) 2)	4.34	***	68.9	***	34.6	***
$\beta_{1,3k}$ (ns(DIM) 3)	13.8	***	86.3	***	-	-
$\beta_{1,4k}$ (ns(DIM) 4)	−13.5	***	99.7	***	-	-
$\beta_{2,1k}$ (AFC medium)	−0.268	n.s.	−3.64	n.s.	−21.3	***
$\beta_{2,2k}$ (AFC high)	1.24	*	50.9	***	−24.8	***
$\beta_{3k}$ (SEAS ^8^ warm)	−0.021	n.s.	0.918	n.s.	−9.03	*
** *Random* **	**SD ^9^**		**SD**		**SD**	
$b_{i0k}$ (intercept)	4.61		48.3		129	
$b_{i1,1k}$ (ns(DIM) 1)	7.56		43.3		191	
$b_{i1,2k}$ (ns(DIM) 2)	6.64		44.8		106	
$b_{i1,3k}$ (ns(DIM) 3)	11.4		74.5		-	
$b_{i1,4k}$ (ns(DIM) 4)	10.1		45.4		-	

*** *p* < 0.001, ** *p* < 0.01, * *p* < 0.05, n.s. *p* ≥ 0.05. ^1^ Mean estimate of the effect; ^2^ observed level of significance; ^3^ age at first calving; ^4^ milk yield (kg/d); ^5^ body weight (kg/d); ^6^ rumination time (min/d); ^7^ natural spline of days in milk (1, 2, 3, 4: sub-polynomials); ^8^ season of the recording period; ^9^ standard deviation.

**Table 4 animals-12-03494-t004:** Predictive accuracy measures of the algorithm.

DIM ^1^	Farm	AUC ^2^		PE ^3^	
		$t_{1}$ ^4^	$t_{2}$ ^5^	$t_{1}$	$t_{2}$
60	Italian	0.558	0.505	0.098 ^†^	0.263
	Belgian 1	0.580 *	0.497	0.091 ^†^	0.228 ^†^
	Belgian 2	-	0.451	-	0.146 ^†^
	British 1	0.526	0.519	0.061 ^†^	0.202 ^†^
	British 2	-	0.498	-	0.196 ^†^
	British 3	0.476	0.508	0.037 ^†^	0.164 ^†^
150	Italian	0.605	0.556	0.096 ^†^	0.256
	Belgian 1	0.578	0.513	0.085 ^†^	0.225 ^†^
	Belgian 2	-	0.475	-	0.143 ^†^
	British 1	0.562	0.526 *	0.060 ^†^	0.202 ^†^
	British 2	-	0.520	-	0.194 ^†^
	British 3	0.535	0.514	0.036 ^†^	0.164 ^†^
240	Italian	0.616	0.566 *	0.096 ^†^	0.259
	Belgian 1	0.597 *	0.533	0.083 ^†^	0.229
	Belgian 2	-	0.507	-	0.143 ^†^
	British 1	0.577	0.539	0.060 ^†^	0.200 ^†^
	British 2	-	0.593 *	-	0.189 ^†^
	British 3	0.763 *	0.559 *	0.035 ^†^	0.158 ^†^

* Significantly higher than 0.5; ^†^ significantly lower than 0.25. ^1^ Days in milk of recorded sensor data to obtain predictions; ^2^ mean Area Under the Curve over 9 cross-validation runs; ^3^ mean error of prediction over 9 cross-validation runs; ^4^ average second calving time; ^5^ average third calving time.

## Data Availability

The data presented in this study are available upon request from the corresponding author. The data are not publicly available due to privacy.

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
