# Peer review of "Joint Models to Predict Dairy Cow Survival from Sensor Data Recorded during the First Lactation"

_animals, 2022, doi:10.3390/ani12243494_

Round 1

Reviewer 1 Report

General comment

Application of joint modeling in livestock survival data is a relatively less research area. The current manuscript presented the joint modeling of longitudinal behavioral and production data of cows to predict the culling probability in 2nd and 3rd lactation. Considering the availability of sensor data at dairy farms, the use of such tools to support decision making would be a smart approach. The manuscript appears to target specific professionals (big data analysts); however, the information is important enough to be shared with the readers of the Animals journal. Some specific comments are listed below:

Line 22-23: Make the first sentence simple and easy to understand. Rest of the summary is fine.

Line 55-56: Remove the information in the parentheses. It could confuse the reader.

Line 83-85: The description of Rizopoulos’s document is making it hard to understand. Please remove these lines. This can be presented in materials and methods section. The information on longitudinal data collected during lactation would be great. Something like this “The milk yield and sensor data are collected longitudinally at farms and could be used in joint modeling”. The authors are requested to rephrase it accordingly.

Line 89-94: This is the objective of the study and should be the last paragraph of the introduction section.

Line 95-99: It is the justification of using joint models in survival analysis. It should come before the objective of the study, preferably at the end of the 3rd paragraph while explaining the use of joint models.

Overall, the introduction was well constructed.

Line 107: I believe body weight change would have been a strong predictor instead of BW. It would be great if the authors use the body weight change during early lactation and use it as a predictor for survival in next lactation. The change could be weekly or fortnightly.

Line 144: At other places the sensor data was used for 240 days. Please clarify this.

Line 153: This subheading “Algorithm development” has a lot of exact text from the Dimitris’s R document. Please be careful about the similarity of the text.

line 218: Please add some detail of 3-fold validation. It would increase the understanding of the text for the readers with little or no knowledge about machine learning.

Line 221: Using alternate groups as a testing set? If so, then please elaborate it by adding some text.  

Line 249: PE? Describe the complete words.

Line 256: Please avoid the repetition of methodology in results section. Please describe only the findings in the results section and discuss these in discussion section.

Line 264: Would it be possible to present the figures with variables (MY, BW, RUM) as a univariate and then in a joint model to show the change in survival probability?

Line 316: Discussion was fine considering the use of the survival prediction modeling in livestock studies.

Line 372: Conclusion was related to main findings of the study.

Reviewer 2 Report

General comment:

An analysis of cow survival using longitudinal data makes it possible to examine the same cows at different time moments, creating a set of dependent observations. The use of joint models for longitudinal and survival data is especially significant for many clinical studies, e.g. on cancer, and can offer better possibilities for predicting survival time (by taking into account relationships and associations between longitudinal and time-to-event data). These methods allow for including two data types into one model. Therefore, it is possible to draw more precise conclusions about the temporal effects of different factors. Their application to the survival analysis in cows is quite an innovative aspect, making it possible to better estimate the effect of different factors on time to culling, and decreasing estimation error at the same time.

Detailed suggestions:

Some minor corrections of the English language are required.

l. 69: “So far, not many attempts have been made to predict dairy cow survival”. Do you really think the number of studies on dairy cow survival prediction is too small?

l. 90: Please explain your selection of variables (MY, BW and RUM), especially RUM (some additional references would be necessary). Would the selection of other traits potentially result in better outcomes?

l. 125: Why did you express AFC as a categorical variable?

l. 146: Why only 305 days and not the complete lactations?

l. 150: Please provide some basic statistics such as means and SD.

l. 159: Does a subject mean a cow?

l. 164: Please provide a more detailed explanation of the symbols used in the equations (to which effects do they really refer?). The current explanation is too general.

l. 218-223: A reference to CV would suffice. There is no need to describe it thoroughly.

l. 232: Please explain the estimation procedure more clearly.

l. 316: The results of the study are, in general, discussed in the Discussion section, but it would be more convincing to compare them with those obtained with a different model (e.g. only the Cox proportional hazards model). In such a case, the results obtained by the authors (even if not the best ones), would clearly indicate that the joint models are a more promising approach than other methods applied to the same dataset.

l. 353: Please explain the way in which you would include the cause of death of the cows as a predictor in the models.

\

.

Reviewer 3 Report

The manuscript is well structured, well-written and relatively easy to follow. The object of the research is focused enough and well defined. The proposed methodology is of interest (especially the joint modeling approach) and could be applied to other cow modeling problems and datasets. I do not have suggestions to improve the quality of the manuscript.

Author Response

The Authors wish to thank the Reviewer for his/her positive comment.

Round 2

Reviewer 1 Report

The authors adequately addressed the technical comments. Some minor edits in the introduction section are suggested below.

Line 74: I rearranged the text from Lines 74-96. Replace the lines with the following edited text.

 “These measurements can be used to predict a cow survivability using new statistical methods. These methods are based on the joint modelling of longitudinal and time-to-event data [15]. The joint models are especially used in the field of biomedicine to predict patients’ survival probabilities based on temporal trajectories of a disease specific biomarkers, and to discriminate between patients with low or high risk of mortality. These models are versatile, being easily adaptable to different recording periods of longitudinal data, time points of survival prediction, and variables to be used in the model. Furthermore, these models avoid deriving biologically meaningful proxies from time-series data, since these directly estimate the information provided by the raw (nearly unprocessed) longitudinal data.

                The aim of the present study was to explore the adoption of a joint model that used first lactation longitudinal sensor data of milk yield (MY), body weight (BW), and rumination time (RUM) to predict cows’ survival to the subsequent lactations.”

Author Response

The Authors replaced the lines rearranged by the Reviewer.

We wish to thank the Reviewer for the editing.